# Respiration aligns perception with neural excitability

**Daniel S Kluger[1,2]\*, Elio Balestrieri[2,3], Niko A Busch[2,3], Joachim Gross[1,2,4]**

[1]Institute for Biomagnetism and Biosignal Analysis, University of Münster, Münster, Germany; [2]Otto Creutzfeldt Center for Cognitive and Behavioral Neuroscience, University of Münster, Münster, Germany; [3]Institute of Psychology, University of Münster, Münster, Germany; [4]Centre for Cognitive Neuroimaging, Institute of Neuroscience and Psychology, University of Glasgow, Glasgow, United Kingdom

**Abstract** Recent studies from the field of interoception have highlighted the link between bodily and neural rhythms during action, perception, and cognition. The mechanisms underlying functional body-brain coupling, however, are poorly understood, as are the ways in which they modulate behavior. We acquired respiration and human magnetoencephalography data from a near-threshold spatial detection task to investigate the trivariate relationship between respiration, neural excitability, and performance. Respiration was found to significantly modulate perceptual sensitivity as well as posterior alpha power (8–13 Hz), a well-established proxy of cortical excitability. In turn, alpha suppression prior to detected versus undetected targets underscored the behavioral benefits of heightened excitability. Notably, respiration-locked excitability changes were maximized at a respiration phase lag of around −30° and thus temporally preceded performance changes. In line with interoceptive inference accounts, these results suggest that respiration actively aligns sampling of sensory information with transient cycles of heightened excitability to facilitate performance.

## Editor's evaluation

Kluger and colleagues investigated the influence of respiration on visual sensory perception in a near-threshold task and argue that the detected correlation between respiration phase and detection precision is liked to α power, which in turn is modulated by the phase of respiration. The main finding is that the moment-to-moment relationship between excitability and perception in turn is coupled to the body's slower respiratory oscillation. This advances our understanding of how the brain-body system works as a whole.

**\*For correspondence:** daniel.kluger@uni-muenster.de

**Competing interest:** The authors declare that no competing interests exist.

## Introduction

Human respiration at rest is a continuous, rhythmic sequence of active inspiration, and passive expiration (*Fleming et al., 2011*). Even the active phase seems effortless, however, thanks to medullar microcircuits like the preBötzinger complex who autonomously and periodically drive respiration (*Moore et al., 2013*). Despite being largely automatic, breathing can also be adapted top-down when required, for example, during speech or laughter (*McKay et al., 2003*). To this end, intricate connections from key structures like the preBötzinger complex (through the central medial thalamus) and the olfactory bulb (OB) to both the limbic system (*Carmichael et al., 1994*) and the neocortex (*Yang and Feldman, 2018*) serve the bidirectional interplay between respiratory control and higher cognitive functions. In turn, neural oscillations in the cortex have long been established as sensitive markers of brain states (*Thut et al., 2012*), which has brought increasing attention to questions of respiration-brain coupling. While there is a rather extensive body of literature on the interaction

of respiratory and neural rhythms in animals, these links are only beginning to be addressed in the human brain. Recently, both invasive (*Zelano et al., 2016*) and noninvasive work (*Kluger and Gross, 2021*) has shown respiration to modulate neural oscillations across a wide network of cortical and subcortical areas, including those not typically associated with olfaction. Moreover, modulatory effects of respiration have been demonstrated in motor (*Kluger and Gross, 2020*; *Rassler and Raabe, 2003*) as well as in cognitive tasks (*Perl et al., 2019*; *Zelano et al., 2016*). Taken together, these studies provide strong evidence for breathing-related changes in neural signaling that, particularly in task contexts, translate into changes in behavior. Yet, fundamental questions in the trivariate relationship between respiration, neural oscillations, and behavior remain unanswered. Respiration-locked behavioral changes have so far been demonstrated in 'higher-level' cognitive paradigms (e.g., visuospatial rotation or emotional judgment tasks), which complicates the identification of a clear mechanism by which respiration shapes perception and behavior. Here, we used a simple perception task to propose neural excitability as a key moderator underlying behavioral changes coupled to respiration phase.

Cross-frequency phase-amplitude coupling is a well-established mechanism of neural information transfer across spatiotemporal scales (*Canolty and Knight, 2010*) and provides an intuitive explanation how the link between respiration phase and oscillatory amplitudes is potentially implemented: In mice, slow respiration-induced rhythms within the OB were shown to be transmitted through the piriform cortex and subsequent cortico-limbic circuits to modulate the amplitude of faster oscillations in upstream cortical areas (*Fontanini and Bower, 2006*; *Ito et al., 2014*). In order to plausibly explain the perceptual and behavioral modulation effects in humans, the respiratory rhythm has to be coupled to specific neural rhythms that code transient brain states of heightened susceptibility for sensory stimulation. These phasic cycles of neural excitability determine the intensity (or *gain*) of early sensory responses in order to amplify relevant or salient stimuli at the expense of irrelevant ones (*Morillon et al., 2015*; *Schroeder et al., 2010*). Excitability has been shown to vary over the respiration cycle in rats (*Dulla et al., 2005*) and is tightly coupled to the amplitude of human cortical alpha oscillations (8–13 Hz): Particularly in the visual system, prestimulus alpha power over parieto-occipital sensors is inversely related to early visual responses (*Chapeton et al., 2019*; *Iemi et al., 2019*; *Romei et al., 2010*). As a consequence, one highly replicated finding is that a prestimulus decrease in alpha power leads to increased detection rates for near-threshold stimuli (*Ergenoglu et al., 2004*; *Iemi and Busch, 2018*; *Iemi et al., 2017*). Coming back to respiration-brain coupling and excitability, not only do performance levels in cognitive tasks fluctuate over the respiration cycle (*Perl et al., 2019*), but so does spontaneous alpha activity (*Kluger and Gross, 2021*). Therefore, respiration-entrained fluctuations in neural excitability represent a promising mechanism for unifying neural and behavioral findings. Both theoretical accounts and evidence from animal and human studies strongly suggest that this link is by no means accidental, but an example of *active sensing* (*Schroeder et al., 2010*). While it is admittedly easier to imagine the high-frequency sniffing and whisking of mice as an active sampling of sensory stimuli, a similar case can be made for human respiration from the perspective of predictive processing (*Mumford, 1992*): In temporally coordinating the breathing act and internal brain dynamics (i.e., heightened excitability), the sampling of bottom-up sensory information can be aligned with top-down predictive streams (*Arnal and Giraud, 2012*). This active view of respiration is corroborated by reports that human participants spontaneously inhale at trial onset in a self-paced cognitive task (*Perl et al., 2019*), effectively aligning stimulus processing with the inspiration phase.

Overall, evidence from the lines of research examined so far point to respiration as a mechanism to synchronize perception and neural excitability, which in turn might influence human performance in a variety of tasks. However, this mechanism currently lacks a clear demonstration, which we aim to provide with the present experiment. We simultaneously recorded respiration, whole-head magnetoencephalography (MEG), and performance measures from 30 human participants in a spatial detection task to address the questions introduced earlier: First, we investigated whether respiration cyclically affects sensitivity toward near-threshold stimuli and how this link develops over the respiration cycle. Second, using parieto-occipital alpha power as a well-established proxy of excitability, we assessed respiration-locked excitability changes and how they are related to performance. Finally, we aimed to illuminate the overarching link between respiration-excitability and respiration-performance modulations.

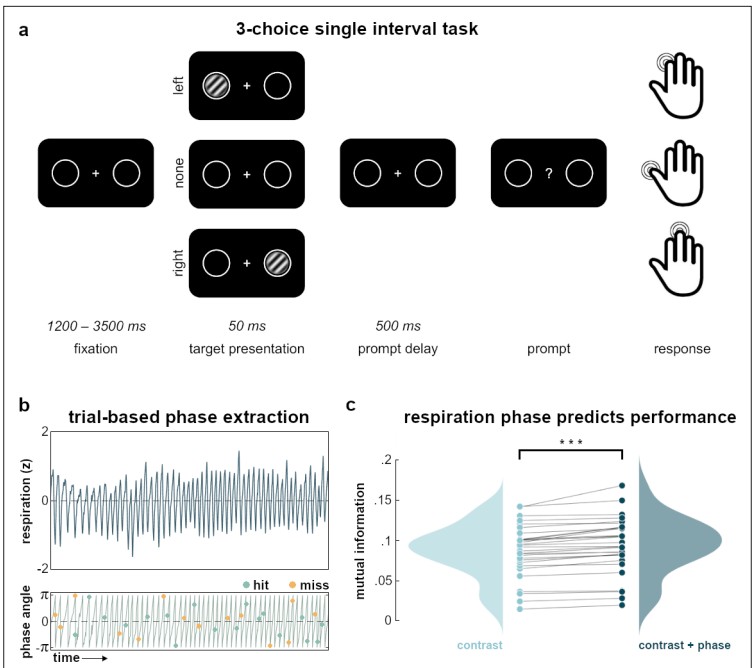

**Figure 1.** Task and behavioral results. (**a**) In the experimental task, participants kept their gaze on a central fixation cross while a brief, near-threshold Gabor patch (magnified for illustrative purposes) was randomly presented either on the left or the right side of the screen. During catch trials, no stimulus was presented. After a brief delay, participants were prompted to indicate whether they had seen a stimulus on the left (index finger) or on the right side (middle finger), or no stimulus at all (thumb). (**b**) Exemplary segment of respiration recordings (normalized, top) plotted against its phase angle (bottom). Over time, targets were randomly presented over the respiration cycle and could be either detected (*hits*, green) or undetected (*miss*, yellow). (**c**) Raincloud plots show mutual information between task performance and stimulus contrast (left) and stimulus contrast plus respiration phase (right), respectively. Each dot for *contrast* or *contrast+ phase* represents one participant's mutual information value computed across all trials for the respective condition. Links between dots identify mutual information values from the same participant and illustrate the increase in mutual information when respiration phase is included in its computation: Mutual information was significantly enhanced by adding respiration phase (Wilcoxon signed-rank test: z=4.78, p<0.001). Filled curves represent the respective probability density functions of mutual information values across participants. Raincloud plots generated with the *RainCloudPlots* toolbox for Matlab (***Allen et al., 2019b***).

The online version of this article includes the following figure supplement(s) for figure 1:

**Figure supplement 1.** Group-level distribution of individual hit rates across all trials.

## Results

### Respiration modulates perception

Participants performed a simple detection task in which Gabor patches were briefly presented at near-threshold contrast either to the left or to the right of a central fixation cross. After a short delay, participants were to report via button press whether they had seen the target on the left, the right, or no target at all (see ***Figure 1a*** and Materials and methods for details). An adaptive staircase (QUEST) was used to adjust the contrast of target trials in a way that performance would settle at a hit rate (HR) of around $\mu_{HR}=0.60$. Across participants, we observed an average HR (i.e., the proportion of detected targets) of $M_{HR}=0.54\pm0.05$ (M ± SD), which was reasonably close to the staircase's desired HR.

While individual HRs were obviously dependent on the contrast of each target, one of our main aims was to assess the independent influence of the interoceptive breathing signal. As a first analysis of potential respiration-related performance changes, we employed a model comparison of single-trial mutual information. For each participant, we first computed mutual information between the discrete factor detected/undetected and the continuous factor of Copula-normalized stimulus contrast. In a second computation, we included another continuous factor with sine and cosine of

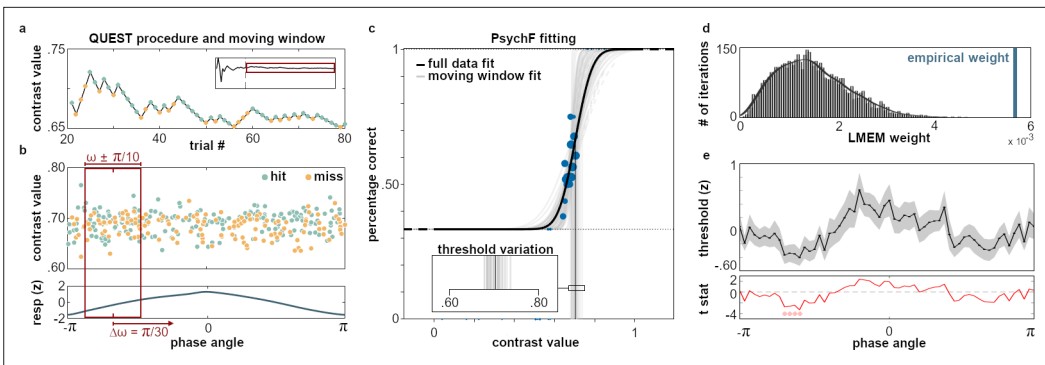

**Figure 2.** Respiration phase modulates performance. (**a**) The first 20 trials of each run were discarded to allow initiation of the QUEST staircase (inset shows contrast variation of the full run). Extending the behavioral analysis shown in *Figure 1*, we obtained contrast values and respiration phase angle of hits (green) and misses (orange). Catch trials did not inform the QUEST staircase and are not shown. (**b**) Top panel shows the contrast values of all hit and miss trials of one exemplary participant sorted by the respiration phase angle at which the targets were presented. Moving along the respiration cycle (bottom) in increments of π/30, we applied a moving window with a width of π/5 to select a subset of trials for a phase-locked refitting of the psychometric function. Phase angles from −π to 0 covered inspiration whereas the range from 0 to π was expiration. (**c**) The psychometric function was first fitted to the full data set (black function, shown here for one typical participant). All parameters except the threshold were then fixed and used as priors for fitting the psychometric function iteratively to an angle-specific subset of trials (gray functions). Blue dots mark clusters of single trials scaled by frequency of occurrence. For each participant, the procedure of trial selection and refitting was repeated 60 times until all trials from all respiration phase angles had been sampled (see panel (**b**)). Vertical gray lines indicate the respective thresholds of refitted psychometric functions, whose variation is shown in the inset. (**d**) Histogram shows the empirical LMEM weight for the respiratory vector norm against the null distribution computed from 5000 iterations of randomized PsychF vectors on the subject level. (**e**) The moving window approach resulted in individual courses of PsychF threshold over respiration angles whose normalized mean is shown here ± SEM (top panel, black line, and shading). These modulations corresponded to contrast value changes within the range of [0.64–0.80] (5th and 95th percentile, respectively). Bottom panel shows t values from cluster-permutation testing. On the group level, PsychF threshold was significantly lowered during the inspiratory phase (covering a respiration phase angle between −128° and −76°, see main text).

The online version of this article includes the following figure supplement(s) for figure 2:

**Figure supplement 1.** Individual psychometric functions (PsychF), fitted on the full data set.

**Figure supplement 2.** Individual PsychF thresholds, iteratively refitted across respiration phase bins.

the respiration phase angle at which each trial was presented (see *Figure 1b* for an illustration). On the group level, we then compared the mutual information with and without respiration phase by testing participant-level differences against zero. A Wilcoxon signed-rank test confirmed that mutual information was significantly enhanced by adding respiration phase information (z=4.78, p<0.001; see *Figure 1c*). Having provided first evidence for a functional link between respiration and perceptual performance, we next conducted an in-depth analysis to illustrate how such behavioral modulation occurs independently of target contrast.

## Respiration changes the psychometric function

Since the previous analysis had established a significant relationship between respiration and perceptual performance, we next addressed the question if this relationship is mediated by a respiration-induced change of the individual psychometric function that relates stimulus contrast to perceptual performance.

In order to assess respiration-related changes in perceptual accuracy irrespective of QUEST-induced changes in stimulus intensity, we exploited single-trial information about both stimulus (i.e., its contrast) and performance (i.e., whether it was detected or not) in an iterative refitting of the psychometric function: For each participant, we first fitted an overall psychometric function to all target trials. The resulting parameters (threshold, width) were used as priors for a moving window approach in which we iteratively refitted the psychometric function to a subset of trials presented at

a certain range of respiration angles (see description above and *Figure 2b*). For each of the 60 over-lapping phase angle bins ranging from $-\pi$ to $\pi$, we thus obtained a threshold estimate characterizing perceptual performance at that respiration phase: With a fixed response criterion of $\mu_{HR}$=0.60, a lower threshold indicates higher accuracy, as a lower stimulus intensity was sufficient to yield the same level of performance. Indeed, individual thresholds showed systematic variations across the respiratory cycle (*Figure 2c*), indicating that respiration in fact changes the perceptual threshold. We set up a linear mixed effect model (LMEM) expressing the refitted PsychF threshold values as a linear combination of sine and cosine of the respiration signal. The resulting regression weights for sine and cosine were combined in a phase vector norm (akin to a harmonic regression) and tested for significance using 5000 random permutations of subject-level PsychF threshold courses (see Materials and methods for details). The empirical regression weight was greater than anyone from the randomization distribution (see *Figure 2d*), corroborating the overall influence of respiration on perceptual performance. With regard to the time course of these modulations, cluster-permutation testing confirmed a significant lowering of the PsychF threshold during the inspiratory phase, corresponding to respiration angles between −128° and −76° (*Figure 2e*).

Taken together, our analysis demonstrates that the individual psychometric function changes significantly across the respiratory cycle. The next step was to assess the role of alpha oscillations and how they are linked to performance changes. If respiration, as we propose, does shape perception through changes in cortical excitability, we can form precise assumptions regarding parieto-occipital alpha power as a proxy of excitability.

## Alpha power is related to behavioral performance

For the excitability hypothesis to hold true, changes in pre- and peristimulus alpha power over parieto-occipital sensors would first have to be linked to changes in behavior. Replicating a rich body of previous work (reviewed in *Samaha et al., 2020a*), we compared parieto-occipital MEG power spectra for detected and undetected targets (see *Figure 3a*). Time-frequency analyses confirmed significant suppression of alpha power prior to presentation of targets which were later detected (vs. undetected, see *Figure 3b* for cluster-corrected results across ROI sensors). Prestimulus alpha suppression persisted as long as 1 s prior to target onset (*Figure 3c*) and was mainly localized over occipital and posterior parietal sensors. Whole-scalp control analyses across all frequency bands demonstrated that this topographical pattern was unique to alpha and beta prestimulus power (see *Figure 3—figure supplement 1*): Compared to the clear parieto-occipital topography of prestimulus alpha modulations, delta and theta effects were prominently shifted to anterior electrodes, which rendered their involvement in low-level visual processing unlikely. No significant effects were observed in the gamma range. In contrast, beta-band modulations were closest to the alpha effects in their topography, covering parietal as well as occipital sites. Although the size of normalized effects was markedly smaller in the beta band (compared to alpha frequencies), the topographic distribution of prestimulus modulations as well as the spectral proximity of the two bands prompted further investigation of beta involvement.

In line with previous work (*Wyart and Tallon-Baudry, 2009*; *Iemi et al., 2017*), we further observed a pronounced alpha desynchronization after stimulus onset for detected (vs. undetected) targets. Such differences have been shown to be driven by stronger evoked responses for these stimuli in near-threshold perception tasks (*Busch and VanRullen, 2010*; *Bareither et al., 2014*) and to influence detection performance together with prestimulus alpha fluctuations (*Wyart and Tallon-Baudry, 2009*).

While these results are confirmatory rather than novel, they pose a central prerequisite for a unifying account of excitability. We suggest that respiration cyclically organizes states of cortical excitability, states which in turn determine perceptual performance. Building on our findings that (a) respiration cyclically modulated performance consistently across participants and (b) performance was determined by phasic changes in parieto-occipital alpha power, the final questions thus concerned the overarching link between alpha power, behavior, and respiration.

## Alpha power, behavior, and respiration

Particularly, two key questions remained for the final analyses: First, for excitability to be the mediator of respiration-locked performance changes, parieto-occipital alpha power should itself be modulated by respiration. Second, such respiration-induced changes in alpha power should contribute to the

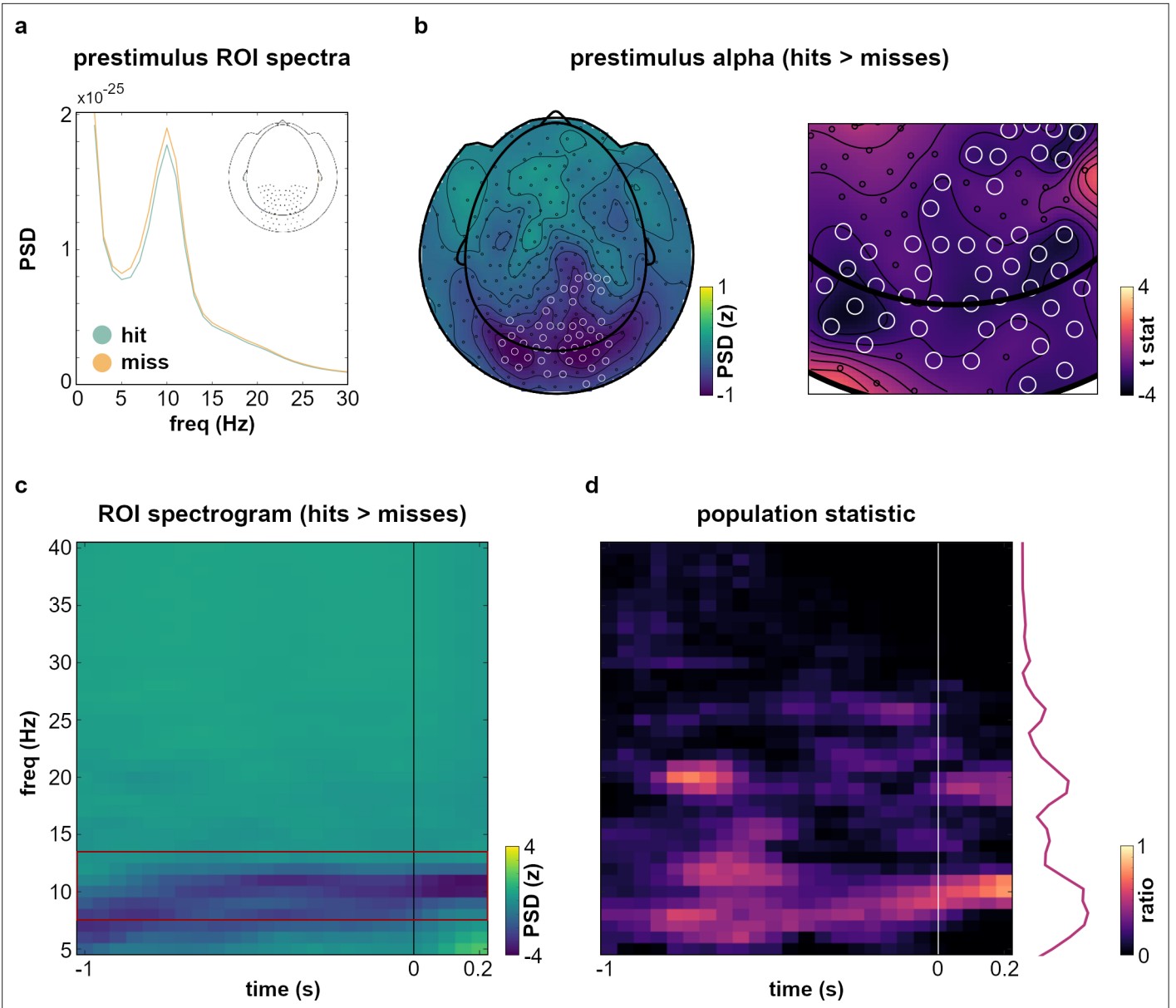

**Figure 3.** Pre- and peristimulus alpha suppression determine perceptual accuracy. (**a**) Frequency spectra show prestimulus alpha suppression (interval [–1 0] s before target onset) for detected (*hits*, green) versus undetected targets (*misses*, yellow) averaged over bilateral parieto-occipital sensors (see inset). (**b**) Left panel shows the topographic distribution of normalized prestimulus power differences between detected and undetected targets (i.e., hits and misses). ROI channels with significant alpha-band differences (computed between [***Arnal and Giraud, 2012***; ***Busch and VanRullen, 2010***] 8–13 Hz) for the contrast hits> misses are marked. Right panel shows the corresponding group-level t values from cluster-corrected permutation testing. Significant channels are marked for reference. (**c**) TFR shows power difference for the contrast hits> misses averaged across ROI sensors. Differences were computed on a frequency range of 5–40 Hz within a time frame of [–1 0.2] s around target onset, red frame indicates alpha frequencies (8–13 Hz). (**d**) Population statistic quantifying TFR differences (see panel (**c**)) on the sensor level. For each frequency at each time point, the heatmap illustrates which ratio of the n=82 ROI channels showed a significant difference for the contrast hits> misses (cluster-corrected, see main text).

The online version of this article includes the following figure supplement(s) for figure 3:

**Figure supplement 1.** Band-specific topographies over time.

phase-dependent performance effects we saw in the PsychF refitting, extending the global differentiation between hits and misses shown above.

As for respiration-induced changes in parieto-occipital alpha power, we replicated previous findings of phase-amplitude coupling in the alpha band (***Kluger and Gross, 2021***). The modulation index (MI; ***Tort et al., 2008***) quantifies the extent to which respiration phase modulated alpha power over

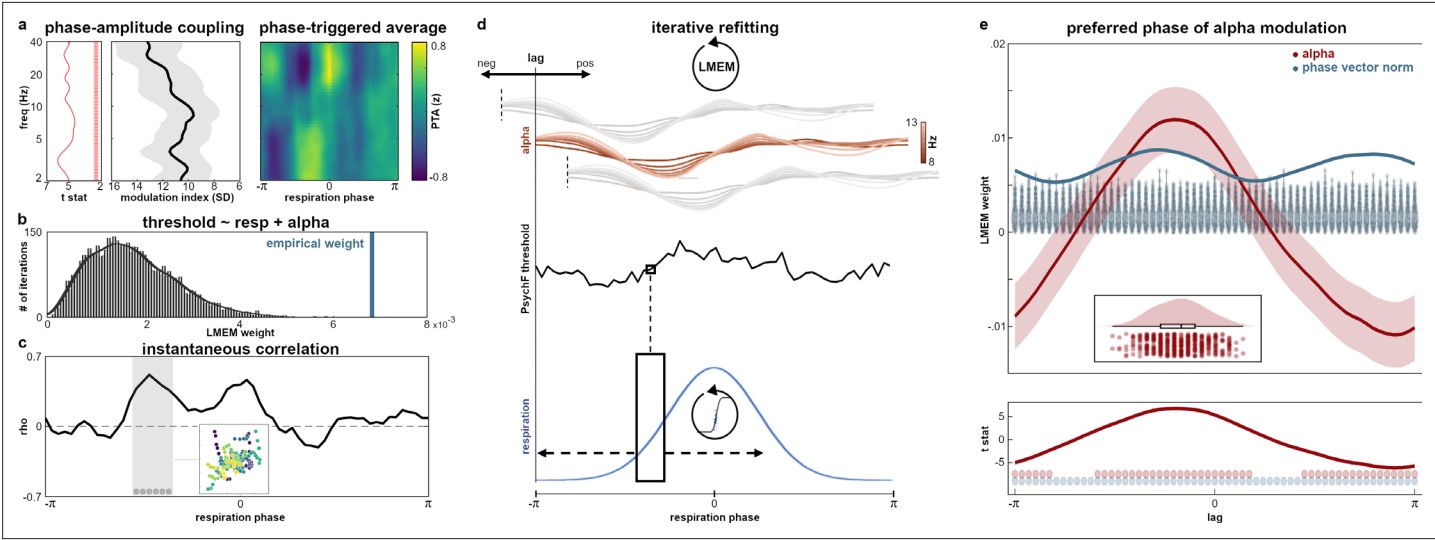

**Figure 4.** Respiration-locked alpha power modulations drive perception effects. (**a**) Left panel shows modulation index as a measure of phase-amplitude coupling across frequencies and corresponding t values from cluster-permutation testing. Coupling was significant (FDR-corrected) over the entire frequency range from 2 to 40 Hz within the parieto-occipital ROI (see *Figure 3a*). Right panel shows ROI sensor-level TFR illustrating group-level average power changes over the course of one entire respiration cycle. Power was normalized within each frequency to highlight phasic changes and is shown in z values. (**b**) Histogram shows the empirical LMEM weight for the respiratory vector norm against the null distribution computed from 5000 iterations of randomized PsychF vectors on the subject level. (**c**) Instantaneous group-level correlation between individual alpha and PsychF threshold courses (averaged between 8 and 13 Hz) with significant phase vector (length of six time points) marked by gray dots (cluster-corrected). Inset shows the correlation within the significant time window: For each participant, six dots represent alpha power and Psych F at the respective significant time point (color-coded across participants). (**d**) Didactic panel illustrating the iterative refitting approach. Individual alpha and PsychF threshold courses (see top and middle) were continuously shifted against each other. At each lag (ranging from −π to π), the LMEM was recomputed on the group level to obtain the preferred phase of alpha modulation effects, that is, the lag at which alpha explained the maximum variance in PsychF thresholds. (**e**) Top panel shows LMEM beta weights ±95% confidence interval for alpha power (red) for all phase lags between alpha and PsychF threshold courses. Alpha effects were found to be strongest at a phase lag of −30°, that is, preceding PsychF modulation by π/6. Blue line shows the respiratory phase vector norm (i.e., the combination of LMEM weights for respiratory sine and cosine). Blue violin plots show the null distributions of the phase vector norm at each lag, resulting from n=1000 random permutations. Inset shows distribution of peak lags for the alpha power coefficient from a bootstrapping approach with n=500 iterations. Bottom panel shows the corresponding t-statistic for alpha across all phase lags. Dots indicate significance for alpha (red) and the respiratory phase vector norm (blue).

The online version of this article includes the following figure supplement(s) for figure 4:

**Figure supplement 1.** Individual phase-triggered averages (top), normalized respiration time courses (plus mean respiration time course in bold; middle), and normalized PsychF threshold over the respiration cycle (bottom).

**Figure supplement 2.** Instantaneous correlation of beta power and perceptual sensitivity.

parietal and occipital electrodes. Depicted in units of standard deviation of the null distribution in *Figure 4a*, values were significant for a wide frequency range including the alpha band (tested against the 95th percentile of a null distribution constructed from 200 random phase shifts, cluster-corrected for multiple comparisons at α=0.05). This result establishes a significant respiratory modulation of the cortical parieto-occipital alpha rhythms that are known to be related to perceptual performance (*Figure 3*).

To assess the significance of alpha contributions to the interplay of respiration and performance, we set up a second LMEM expressing refitted PsychF threshold values as a linear combination of sine and cosine of the respiration signal as well as parieto-occipital alpha power (averaged from 8 to 13 Hz). Using the permutation approach described above, we replicated the significant effect of the respiratory phase vector norm from the first LMEM (all p<0.002, see *Figure 4b*). Moreover, the model yielded a significant effect for parieto-occipital alpha power (t(1794)=6.02, p<0.001). In addition, alpha power significantly interacted with the cosine portion of the respiratory signal (t(1794)=4.07, p<0.001), providing strong support for both respiration and excitability as significant predictors of perceptual performance. Comparing the LMEM to the previous model (which only contained respiratory sine and cosine) confirmed the significant contribution of alpha power in explaining PsychF

threshold variation by means of a theoretical likelihood ratio test ($\chi^2$(3)=49.45, p<0.001). In other words, variation in perceptual accuracy was around 49 times more likely under the model which included parieto-occipital alpha than under one that did not. Of note, the increase in model fit cannot be explained by the higher number of regressors, as the likelihood ratio test accounts for the addition of predictors.

A third LMEM investigating potential beta contributions yielded significant effects for both alpha (t(1790)=3.27, p<0.001) and beta power (t(1790)=4.83, p<0.001). Beta showed significant interactions with the sine of the respiratory signal (t(1790)=–3.52, p<0.001) as well as with alpha power (t(1790)=–4.63, p<0.001). Comparing the LMEM to the previous model which only contained alpha power (along with respiratory sine and cosine) confirmed the significant contribution of beta power in explaining PsychF threshold variation by means of a theoretical likelihood ratio test ($\chi^2$(4)=60.43, p<0.001).

Finally, we aimed to characterize the relationship between alpha power and PsychF threshold over the respiration cycle. To this end, we computed the instantaneous (i.e., bin-wise) correlation between alpha power (averaged between 8 and 13 Hz) and PsychF threshold values. On the group level, we found the two time courses to be significantly correlated during the inspiratory phase between –90° and –60° (cluster-corrected, *Figure 4c*). These positive correlations corroborated our previous results showing epochs of increased performance during the inspiratory phase (*Figure 2d*) as well as the perceptual benefit of suppression effects in the alpha band (*Figure 3b*). Control analyses across all frequency bands yielded a significant instantaneous correlation between PsychF threshold and beta power as well, albeit at a slightly later phase (see *Figure 4—figure supplement 2*). Consistent with the TFR analysis shown above, no significant correlation between oscillatory power and respiration time courses were found for delta, theta, and gamma bands.

While the instantaneous correlation nicely captured interactions between alpha oscillations and PsychF threshold, it was naturally computed at zero lag between the two time courses. Conceivably, however, if respiration-induced excitability changes in fact mediate performance effects, respiration-brain coupling could temporally precede respiration-induced changes in behavior. Within the present analytical framework, this would manifest in an increased effect of alpha power at negative lags. To test this hypothesis, we combined circular shifts of the alpha power time courses with a bootstrapping approach to recompute the second LMEM described above for different lags ranging from -π to π. In short, we shifted the alpha power courses of each participant against their PsychF threshold courses by a single angle bin (i.e., π/30 or 6°), recomputed the LMEM (including coefficients for sine, cosine, and alpha), and stored the beta weights and t values for each predictor (see *Figure 4d* for an illustration). This way, we obtained a distribution of t values as a function of phase shifts which showed that the effect of alpha power peaked not at zero lag, but at a lag of –30° (*Figure 4e*). To test the reliability of this negative peak, we recomputed binwise LMEMs n=500 times, each time randomly drawing 30 participants (with replacement). The bootstrapping procedure yielded a confidence interval of [–33.17 to 29.25] degrees for the peak effect of alpha power. While these results strongly suggest that respiration-alpha coupling temporally precedes behavioral consequences, they do not provide sufficient evidence for a strict causal interpretation (see Discussion).

To summarize, both respiration phase and parieto-occipital alpha power significantly explained individual variation in perceptual sensitivity (PsychF threshold). While parieto-occipital alpha and PsychF threshold were found to be significantly correlated during the inspiratory phase, the effect of alpha power was maximized at a respiration phase lag of around –30°, showing that alpha modulation precedes behavioral modulation.

## Discussion

The overall picture of our findings emphatically underscores the central role of respiration in the temporal synchronization of information sampling to phasic states of heightened neural excitability. Extending previous reports of respiration-locked performance changes, we found respiration to modulate perceptual sensitivity in a near-threshold detection task. Assessing behavioral effects over the continuous respiration signal allowed us to characterize the shape of this modulation over time, demonstrating for the first time that it appears to be more complex than a dichotomy of inspiratory versus expiratory phase. This modulatory pattern raises interesting methodological questions for future research: As the respiratory signal itself is not strictly sinusoidal, respiration-locked performance modulation is not necessarily constrained to a sinusoidal shape. Conceivably, similar to the

heartbeat-evoked potential shown for the cardiac rhythm (*Pollatos and Schandry, 2004*), respiratory rhythms too could trigger event-like behavioral changes that repeat with each breathing cycle. Future studies might investigate if the temporal modulation revealed here is best described by a sinusoidal function spanning the entire breathing cycle or by other models that are possibly restricted to specific time windows within a full cycle.

We then investigated parieto-occipital oscillatory power in the alpha band (8–13 Hz) as a proxy of neural excitability. Alpha power was significantly suppressed prior to detected versus undetected targets, highlighting the behavioral benefits of heightened prestimulus excitability. Furthermore, parieto-occipital alpha was found to be significantly modulated by respiration phase, strongly suggesting a functional coupling of respiration-locked changes in both excitability and performance. This coupling was further accentuated by a significant instantaneous correlation between alpha power and sensitivity, particularly during the inspiratory phase. Notably, the effect of alpha power on behavioral performance was strongest at a respiration phase lag of around –30°, indicating that respiration-alpha coupling temporally precedes performance changes.

## Alpha oscillations, neural excitability, and behavior

There is overwhelming evidence for an intricate connection between alpha oscillations, excitability, and behavior, particularly in the visual domain (see *Samaha et al., 2020a* for a recent review). In addition to alpha pacemaker cells originally shown in the animal thalamus (*da Silva et al., 1973*), widespread alpha activity suggests generators to be further located in early visual (*Bollimunta et al., 2008*) and even higher-order cortical areas (*Florez et al., 2015*; *Haegens et al., 2015*). Excitatory input to the visual cortex is regulated by functional inhibition in a feed-forward mechanism based on alpha oscillations, effectively controlling the excitability of the neural system per se (*Iemi et al., 2021*; *Jensen and Mazaheri, 2010*). The inverse relationship between alpha power and excitability is corroborated by cross-modal evidence that strong ongoing alpha oscillations entail reduced single-unit firing rates in humans (*Chapeton et al., 2019*) and primates (*Dougherty et al., 2017*; *Lundqvist et al., 2020*), population-level activity such as local field potentials (*Potes et al., 2014*; *Spaak et al., 2012*), and hemodynamic BOLD activity (*Becker et al., 2011*; *Goldman et al., 2002*). In perceptual tasks, a transient state of lowered excitability inevitably affects behavior, as evident from longer reaction times (*Kelly and O'Connell, 2013*; *Zhang et al., 2008*), lower confidence reports (*Samaha et al., 2017*; *Samaha et al., 2020b*), and lower detection rates of near-threshold stimuli (*Chaumon and Busch, 2014*; *Iemi and Busch, 2018*). Accordingly, one recent study proposed that the alpha rhythm shapes the strength of neural stimulus representations by modulating excitability (*Iemi et al., 2021*). Previous work by *Michalareas et al., 2016* as well as our own data (see Supplementary Material) point toward an interaction between alpha and beta bands, as beta oscillations have very recently been implicated in mediating top-down signals from the frontal eye field (FEF) that modulate excitability in the visual cortex during spatial attention (*Veniero et al., 2021*). Our findings suggest that this top-down signaling is modulated across the respiration cycle in a way that changes behavioral performance. Thus, our present results not only provide further support for the link between alpha power and excitability-related performance changes, but further suggest respiration as a key contributing factor in the ongoing search for an underlying mechanism. Such an argument for functional body-brain coupling does not contradict the proposed alpha mechanism in any way, but rather broadens the explanatory scope to unify neural and peripheral signaling.

## Excitability changes are coupled to respiration

Compared to the extensive research of alpha oscillations and their relation to neural excitability, the link between excitability and respiration is considerably less well understood. Intracranial recordings in animals have demonstrated that respiration modulates spike rates in a variety of brain regions (*Biskamp et al., 2017*; *Ito et al., 2014*; *Yanovsky et al., 2014*). This characteristic was implemented in a graph model by *Heck et al., 2016*, which provided a proof of principle that sinusoidal input and intrinsic properties of cortical networks are sufficient to potentially achieve respiration-locked modulations of high-frequency oscillations. The authors later proposed two main sources of respiration-locked neural signaling, namely the OB and extrabulbar sources within the brainstem (*Heck et al., 2019*). As outlined in the introduction, there is broad consensus that cross-frequency coupling (*Canolty and Knight, 2010*; *Jensen and Colgin, 2007*) plays a central

role in translating respiratory to neural rhythms: Respiration entrains neural activity within the olfactory tract via mechanoreceptors, after which the phase of this infraslow rhythm is coupled to the amplitude of faster oscillations (see *Fontanini and Bower, 2006*; *Ito et al., 2014*). While this mechanism is difficult to investigate directly in humans, converging evidence for the importance of bulbar rhythms comes from animal bulbectomy studies (*Ito et al., 2014*) and the fact that respiration-related changes in both oscillatory power and behavior dissipate during oral breathing (*Zelano et al., 2016*; *Perl et al., 2019*). Thus, rhythmic nasal respiration conceivably aligns rhythmic brain activity across the brain, which in turn influences behavior. In our present paradigm, transient phases of heightened excitability would then be explained by decreased inhibitory influence on neural signaling within the visual cortex, leading to increased postsynaptic gain and higher detection rates. Given that the breathing act is under voluntary control, the question then becomes to what extent respiration may be actively used to synchronize information sampling with phasic states of heightened excitability.

## Toward an active account of human respiration-brain coupling

Such an active account of respiration-brain coupling is once more motivated by the animal literature. In rodents, respiration is only one of multiple rhythmic processes that constitute orofacial motor behaviors, including head and nose movements, whisking, and sniffing (*Welker, 1964*). Distinct respiratory nuclei in the ventral medulla have been shown to coordinate these complex motor patterns in a specific manner, in that the respiratory rhythm effectively 'resets' faster rhythms like whisking behavior (*Moore et al., 2013*). According to this 'master clock' conceptualization of respiration, respiratory circuits in the brain stem allow phase-locking of various sensorimotor channels through 'snapshots' of the orofacial environment continuously triggered by respiration onsets (*Kurnikova et al., 2017*). In other words, multiple streams of sensory information are coordinated in a way that optimizes their integration and propagation (*Corcoran et al., 2018*). While the concept of active sensing has thus been well established in animals (*Wachowiak, 2011*), interoceptive inference in human sensation is still in its infancy. Intriguingly, theoretical modelling work (*Allen et al., 2019a*; *Corcoran et al., 2018*; *Owens et al., 2018*) as well as empirical studies on various interoceptive signals (*Galvez-Pol et al., 2020*; *Mather and Thayer, 2018*; *Rebollo et al., 2018*) are seeking to widen our understanding of how sampling information from the external world is coordinated with bodily states. Reports on functional body-brain coupling span cyclic signals from infraslow gastric (*Rebollo et al., 2018*) over respiratory (*Herrero et al., 2018*; *Kluger and Gross, 2021*) to cardiac rhythms (*Galvez-Pol et al., 2020*; *Mather and Thayer, 2018*), showing that, across all time scales, sensory information is critically dependent on physiological rhythms. From a predictive processing perspective, the link between interoceptive and neural rhythms serves *predictive timing*, meaning that information sampling itself is temporally aligned with particular bodily states. Using free visual search, *Galvez-Pol et al., 2020* demonstrated that the timing of eye movements was closely coupled to certain phases of the cardiac cycle, with information sampling clustered during quiescent periods of the heartbeat. Functionally, the authors interpret this synchronization to maximize the signal-to-noise balance between interoceptive and exteroceptive signals, which is precisely the mechanism we propose for the alignment with excitability states. A similar argument has prominently been made for attentional selection (*Thut et al., 2006*) which actively phase-locks neural oscillations to sensory streams in order to upregulate response gain and amplify attended stimuli (*Lakatos et al., 2008*). Our findings, together with mounting evidence from recent interoceptive accounts, strongly suggest that well-described accounts of hierarchically organized oscillations (*Lakatos et al., 2005*) are not limited to neural signals alone, but extend to even slower bodily rhythms. Rigorous future work is needed to investigate potentially causal effects of respiration-brain coupling on behavior, for example, by means of directed connectivity within task-related networks. A second promising line of research considers top-down respiratory modulation as a function of stimulus characteristics (such as predictability). This would grant fundamental insights into whether respiration is actively adapted to optimize sensory sampling in different contexts, as suggested by the animal literature. Recognizing human respiration as active sensory selection rather than a mere bottom-up automatism offers a promising framework to explain how respiration-locked changes in neural signaling benefit action, perception, and cognition.

## Materials and methods

### Participants

Thirty volunteers (16 female, age 25.1±2.7 years [mean ± SD]) participated in the study. All participants had (corrected-to) normal vision, denied having any respiratory or neurological disease, and gave written informed consent prior to all experimental procedures. The study was approved by the local ethics committee of the University of Muenster (approval ID 2018-068f-S) and complied with the Declaration of Helsinki.

### Procedure

Participants were seated upright in a magnetically shielded room while we simultaneously recorded respiration and MEG data. MEG data were acquired using a 275-channel whole-head system (OMEGA 275, VSM Medtech Ltd, Vancouver, Canada) and continuously recorded at a sampling frequency of 600 Hz. To minimize head movement, the participant's head was stabilised with cotton pads inside the MEG helmet. Data were acquired across six runs with intermediate self-paced breaks. The length of each run was dependent on individual response speed (452±28 s; M ± SD). Participants were to breathe automatically through their nose while respiration was recorded as thoracic circumference by means of a respiration belt transducer (BIOPAC Systems, Inc, Goleta, USA) placed around their chest. Participants were continuously monitored via video to ensure nasal (and not oral) respiration. To characterize individual rates of natural breathing, we computed TFRs of single-run respiration time courses for each participant. To this end, the continuous respiration signal was segmented into 20 s segments with 50% overlap and subsequently detrended. Power spectra (range 0–2 Hz in 0.025 Hz increments, single Hanning taper, zero-padded to 40 s segments) were computed in Fieldtrip (*Oostenveld et al., 2011*) and subsequently averaged across runs within each participant. On the group level, the dominant breathing frequency was found to be at 0.26±0.08 Hz (M ± SD), corresponding to an average of 15.8 breaths per minute.

### Task

Participants performed a spatial detection task (three-choice, single interval) in which they were to fixate on a cross (0.4° in diameter) in the center of the screen presented against a black background. Following this fixation period (jittered between 1200 and 3500 ms), a small Gabor patch (0.3° in diameter) was presented for 50 ms in a marked circular area (3.5° in diameter) 10° to the left or the right side of the fixation cross. In addition to left or right side targets, one-third of all trials were catch trials where no target was presented at all. After a delay of 500 ms, a question mark in the center of the screen prompted participants to give their response: Using a three-button response box in their right hand, participants reported whether they saw a target on the left (index finger) or the right side (middle finger) or no target at all (thumb). Once the report was registered, a new trial started (again with a fixation period). Participants were instructed to keep their eyes on the center cross at all times and encouraged to report left or right targets even when they were not entirely certain.

For each trial, target contrast was adapted by a QUEST staircase (*Watson and Pelli, 1983*) aimed at individual HRs of about 60%. Each of the six experimental runs contained a total of 720 trials (240 left target, right target, and catch trials, respectively).

### Behavioral analyses

Behavioral data were preprocessed and analyzed using R (*R Development Core Team, 2014*) and Matlab (The Mathworks, Inc, Natick, USA). To account for the initiation of the QUEST procedure, the first 20 trials of each run were discarded. All behavioral analyses were focussed on trials with an actual target present (n=480) which were subsequently classified into hits (i.e., detected targets) and misses (undetected targets). Misses were restricted to 'target absent' responses during target-present trials. Individual HRs were computed as $n_{Hits}/(n_{Hits}+n_{Misses})$.

To obtain the respiration phase angles corresponding to single trials, we used Matlab's *findpeaks* function to identify time points of peak inspiration (peaks) and expiration (troughs) in the normalized respiration time course. Phase angles were linearly interpolated from trough to peak ($-\pi$ to 0) and peak to trough (0 to $\pi$) in order to yield respiration cycles centerd around mean inspiration. For each trial, we thus obtained the respiration angle at which the target had been presented.

Using the *gcmi* toolbox for Matlab (*Ince et al., 2017*), we computed single-trial mutual information on the participant level for two sets of factors: First, we considered a discrete *detection* factor (target detected/undetected) and a continuous *contrast* factor of Copula-normalized target contrast (as determined by the QUEST procedure). The Copula transform is based on rank ordering the individual target contrast vectors (see *Ince et al., 2017* for details). In a second step, we added two further continuous *sine* and *cosine* factors of respiration phase. These computations yielded two mutual information values for each participant, with and without respiration phase, respectively. To determine whether adding respiration phase significantly enhanced mutual information on the group level, we tested individual differences of the two mutual information values against zero with a Wilcoxon signed-rank test.

### Respiration phase-dependent PsychF fitting

For a more sensitive measure of respiration-locked changes in perceptual accuracy, we combined the moving window approach outlined above with a robust Bayesian inference analysis for the psychometric function. For each participant, we first fitted the psychometric function (with a cumulative Gaussian distribution as the sigmoid) to all target trials using a three-alternative forced choice model (guessing rate fixed at 0.33) from the *Psignifit* toolbox for Matlab (*Schütt et al., 2016*). We then used the identified threshold and width (i.e., slope) from the overall fit as priors for an iterative refitting of the PsychF on subsets of trials obtained from the moving window: Again moving along the respiration cycle in increments of $\Delta\omega = \pi/30$, we fitted the PsychF to the trials presented at a respiration angle of $\omega \pm \pi/10$. This way, we extracted respiration phase-dependent estimates of PsychF threshold for each participant (see *Figure 2*) which were subsequently z-scored. Finally, the normalized threshold variations were averaged across subjects to obtain the grand average phase-dependent threshold modulation at the population level.

### MEG preprocessing and segmentation

All MEG and respiratory data preprocessing were done in Fieldtrip for Matlab. Prior to statistical comparisons, we adapted the synthetic gradiometer order to the third order for better MEG noise balancing (*ft_denoise_synthetic*). Power line artifacts were removed using a discrete Fourier transform (DFT) filter on the line frequency of 50 Hz and all its harmonics (including spectrum interpolation; *dftfilter* argument in *ft_preprocessing*). Next, we applied independent component analysis (ICA) on the filtered data to capture eye blinks and cardiac artifacts (*ft_componentanalysis* with 32 extracted components). On average, artifacts were identified in 2.53±0.73 components (M ± SD) per participant and removed from the data. Finally, continuous MEG data were segmented to a [−1.25 to 0.75] s interval around target onset and resampled to 300 Hz. Likewise, respiration time courses were resampled to 300 Hz and z-scored to yield normalized respiration traces for each participant and each experimental run.

### Head movement correction

In order to rule out head movement as a potential confound in our analyses, we used a correction method established by *Stolk et al., 2013*. This method uses the accurate online head movement tracking that is performed by our acquisition system during MEG recordings. This leads to six continuous signals (temporally aligned with the MEG signal) that represent the x, y, and z coordinates of the head center ($H_x, H_y, H_z$) and the three rotation angles ($H_\psi, H_\vartheta, H_\varphi$) that together fully describe the head movement. We constructed a regression model comprising these six 'raw' signals as well as their derivatives and, from these 12 signals, the first-, second-, and third-order non-linear regressors to compute a total of 36 head movement-related regression weights (using a third-order polynomial fit to remove slow drifts). This regression analysis was performed on the power spectra of single-sensor time courses, removing signal components that can be explained by translation or rotation of the head with respect to the MEG sensors.

### MEG time-frequency analyses

Parieto-occipital time-frequency representations (TFRs) of hit and miss trials were computed using a discrete prolate spheroidal sequences multitaper on a frequency range from 5 to 40 Hz (*mtmconvol* argument for *ft_freqanalysis*). We applied a 2-Hz smoothing kernel to frequencies below 30 Hz and a

5-Hz smoothing kernel to frequencies above 30 Hz. As hits were slightly more frequent than misses (recall that the QUEST staircase aimed to fix performance at a HR of 0.60), we randomly sampled hit trials to match the number of misses, thus avoiding sampling bias. TFRs were computed for a moving window of 500 ms (50 ms increments) over the entire time frame of [–1.25 to 0.75] s around target onset. TFRs of hits and misses were separately averaged across participants and collapsed over the time dimension (range [–1 to 0] s) to obtain prestimulus power spectra (see *Figure 3a*).

Again, we used nonparametric permutation testing to determine statistical significance of group-level differences between TFRs of hits and misses. Individual TFRs were represented in matrices of 82 ROI channels × 36 frequencies × 25 time points. Differences across participants were assessed by means of dependent-samples t-tests while correcting for multiple comparisons (across channels, frequencies, and samples) with a Monte Carlo-based cluster permutation approach (5000 randomization iterations, α=0.05).

## Computation of global field power

For the computation of global field power, the time courses of 82 MEG channels within our parieto-occipital ROI were individually subjected to a continuous wavelet transform using a Morlet wavelet with a Fourier-based algorithm for 64 frequencies (ranging from 0.5 to 40 Hz). The Fourier transform of our wavelet is defined as

$$\psi(f) = \pi^{\frac{1}{4}} e^{\frac{-(f-f0)^2}{2}} H(f) \tag{1}$$

where H(f) is the Heaviside function and f0 is the center frequency in radians/sample. Next, we computed the absolute amplitude envelope of this complex-valued data, smoothed it with a 300 ms moving average, and averaged these amplitude values across channels and experimental runs.

## Computation of modulation index and phase-triggered average

The MI quantifies cross-frequency coupling and specifically phase-amplitude coupling. Here, it was used to study to what extent the amplitude of brain oscillations at different frequencies were modulated by respiration phase. To this end, the instantaneous phase of the respiration time course was computed with the Hilbert transform. MI computation was then based on the average oscillatory amplitude (ranging from 2 to 40 Hz) at 20 different phases of the respiratory cycle (centerd around peak inspiration). Any significant modulation (i.e., deviation from a uniform distribution) is quantified by the entropy of this distribution. To account for frequency-dependent biases, we followed previously validated approaches (*Zelano et al., 2016*) and computed 200 surrogate MIs using random shifts of respiratory phase time series with concatenation across the edges. The normalized MI was computed by subtracting, for each frequency, the mean of all surrogate MIs and dividing by their standard deviation leading to MI values in units of standard deviation of the surrogate distribution. The computation resulted in normalized MI values for each channel, frequency, and participant. Group-level significance of these normalized MI values was determined by means of cluster-based permutation testing using *ft_freqstatistics* in Fieldtrip. Specifically, we conducted a series of one-tailed t-tests of individual MI values at each frequency against the 95th percentile of the null distribution from the 200 surrogate MI values. T values were then thresholded at p=0.05 and adjacent significant data points were defined as clusters. Using 5000 randomization iterations and the cluster sum criterion, the original cluster statistics were compared to the histogram of the randomized null statistics. Clusters in the original data were deemed significant when they yielded a larger test statistic than 95% of the randomized null data.

To assess oscillatory modulation over time, the phase-triggered average (PTA) was computed from the smoothed, band-specific amplitude envelopes averaged across the 80 ROI sensors. Time points of peak inhalation were detected from the respiration phase angle time series using Matlab's *findpeaks* function. For each peak in the respiration signal, global field power across all 64 frequencies was averaged within a time window of ±1000 samples centered around peak inhalation. The resulting 64 frequencies × 2000 samples matrix was finally z-scored across the time dimension, leading to normalized oscillatory power within the ROI phase-locked to the respiration signal. This analysis is equivalent to a wavelet-based time-frequency analysis. Computations were done separately for the six experimental runs and subsequently averaged across runs and participants.

## Instantaneous correlation

We investigated the presence of an instantaneous, positive group-level correlation between variations in prestimulus alpha power and variations in the PsychF threshold, both as a function of the respiration phase angle. First, we averaged the alpha power over the range between 8 and 13 Hz. Then, for each phase angle bin, we correlated the alpha power with the corresponding threshold values. This yielded a course of Pearson r and corresponding p values over the respiration cycle. In order to correct for multiple comparisons, we applied a cluster permutation approach as follows: We first detected the significant clusters in our data and obtained the cluster masses by summing together the r values of the contiguous significant points (p<0.05). Then we reiteratively shuffled (N=10,000), the participant order in both the matrices of alpha power and thresholds. For each iteration, we stored the highest cluster mass value, yielding a surrogate distribution of cluster masses. We rejected H0 for each cluster mass in the real data that exceeded the 95th percentile ($\alpha$=0.05, one-tailed) of the surrogate distribution of cluster masses.

## Linear mixed effect modelling

We employed LMEM to investigate the relationship between respiration phase, parieto-occipital alpha power, and perceptual accuracy. LMEM models a response variable (in our case, the PsychF threshold value) as a linear combination of fixed effects shared across the population (phase angle, alpha power) and participant-specific random effects (i.e., modulatory variation between participants). We specified two LMEMs: The first model was set up to predict the PsychF threshold as a function of respiration angle (with separate sine and cosine contributions):

$$T_j = \beta_0 + (S_{1j} + \beta_1) * sin_{resp} + (S_{2j} + \beta_2) * cos_{resp} + e_j \tag{2}$$

For participant j, the PsychF threshold value is expressed as a combination of the intercept ($\beta_0$), the fixed effects of sine and cosine of the respiration angle ($\beta_1$, $\beta_2$), and an error term ($e_j \sim N(0,\sigma^2)$). Since we had observed considerable between-participant variation of respiration-locked PsychF changes during the previous analysis (see above), we specified the current LMEM to include random slopes ($S_{1,j}$, $S_{2,j}$). This accounts for the fact that a potential fixed effect of respiration phase does not necessarily have to modulate PsychF threshold identically across the entire cohort. For significance testing, LMEM beta weights for sine and cosine were combined in a respiratory phase vector norm:

$$v = \sqrt{\beta_2^1 + \beta_2^2} \tag{3}$$

This empirical combination of sine and cosine was tested against a null distribution constructed from randomized individual PsychF courses. To this end, we recomputed the LMEM n=5000 times while shuffling each participant's PsychF course. The resulting beta weights for sine and cosine were combined as described above and saved as 'null vector norms.' Significance of the empirical vector norm was determined by computing its percentile rank relative to the density function of the null distribution.

An analogous approach was used in a second model that further included the fixed effect of alpha power and separate interaction terms:

$$T_j = \beta_0 + (S_{1j} + \beta_1) * sin_{resp} + (S_{2j} + \beta_2) * cos_{resp} + (S_{3j} + \beta_3) * alpha + (S_{4j} + \beta_4) * sin_{resp} * alpha + (S_{5j} + \beta_5) * cos_{resp} * alpha + e_j \tag{4}$$

We used Matlab's *compare* function to test whether including alpha power in the model would significantly increase the LMEM fit (while simultaneously accounting for the addition of a third fixed effect) by means of a theoretical likelihood ratio test.

Next, we attempted to characterize potential phase shifts between the modulatory effects of respiration on PsychF thresholds and alpha power. To this end, for lags ranging from $-\pi$ to $\pi$, we shifted the alpha power course of each participant against their PsychF threshold course by a single angle bin (i.e., $\pi$/30 or 6°), recomputed the group-level LMEM (including coefficients for sine, cosine, and alpha), and stored the beta weights and t values for each predictor. This way, we obtained a distribution of

t values as a function of phase shifts between alpha and PsychF time courses. In keeping with our previous LMEM analyses, we also generated a null distribution for the empirical phase vector norm at each lag by randomly shuffling each participant's PsychF threshold vector n=1000 times. These null distributions are shown as violin plots in *Figure 4e*.

To quantify the reliability of the ideal phase lag we obtained at around –30° (*Figure 4e*), we combined the previous permutation procedure with a bootstrapping approach: For n=500 iterations, we randomly selected 30 participants (with replacement) and recomputed binwise LMEMs as described above.

## Control analyses for extended frequency range

In order to characterize our central findings across an extended frequency range, we conducted control analyses that were not restricted to alpha frequencies alone, but involved all major frequency bands.

To this end, we extended the spectral range for both MEG time-frequency analyses and the computation of global field power (see above) to a range of 2–70 Hz. Computations and settings remained otherwise identical to our main analyses as described above. We then first computed cluster-corrected whole-scalp topographies for delta (2–4 Hz), theta (4–7 Hz), alpha (7–13 Hz), beta (14–30 Hz), and gamma bands (30–70 Hz) for hits versus misses over time intervals 1 s prior to stimulus presentation. Next, we computed the instantaneous correlation between individual beta power (over the respiration cycle) and respiratory phase, analogous to our main analysis shown in *Figure 4c*. Finally, we recomputed the LMEM visualized in *Figure 4* with an additional factor for beta power and corresponding interactions:

$$T_j = \beta_0 + (S_{1j} + \beta_1) * sin_{resp} + (S_{2j} + \beta_2) * cos_{resp} + (S_{3j} + \beta_3) * alpha + (S_{4j} + \beta_4) * sin_{resp} * alpha + (S_{5j} + \beta_5) * cos_{resp} * alpha + (S_{6j} + \beta_6) * beta + (S_{7j} + \beta_7) * sin_{resp} * beta + (S_{8j} + \beta_8) * cos_{resp} * beta + (S_{9j} + \beta_9) * alpha * beta + e_j \tag{5}$$

# Acknowledgements

The authors would like to thank Karin Wilken, Ute Trompeter, and Hildegard Deitermann for their invaluable assistance during data collection. This work was supported by the Interdisciplinary Center for Clinical Research (IZKF) of the medical faculty of Münster (Gro3/001/19). NAB (BU2400/9-1) and JG (GR 2024/5-1) were further supported by the DFG.

# Additional information

### Funding

| Funder | Grant reference number | Author |
|---|---|---|
| Interdisciplinary Center for Clinical Research, University of Münster | Gro3/001/19 | Joachim Gross |
| Deutsche Forschungsgemeinschaft | GR2024/5-1 | Joachim Gross |
| Deutsche Forschungsgemeinschaft | BU2400/9-1 | Niko A Busch |

The funders had no role in study design, data collection and interpretation, or the decision to submit the work for publication.

### Author contributions

Daniel S Kluger, Conceptualization, Data curation, Formal analysis, Investigation, Methodology, Visualization, Writing – original draft, Writing – review and editing; Elio Balestrieri, Formal analysis, Methodology, Visualization, Writing – original draft, Writing – review and editing; Niko A Busch, Conceptualization, Funding acquisition, Methodology, Resources, Software, Supervision, Validation, Writing – review and editing; Joachim Gross, Conceptualization, Formal analysis, Funding acquisition,

Methodology, Project administration, Resources, Software, Supervision, Validation, Writing – review and editing

### Author ORCIDs
Daniel S Kluger http://orcid.org/0000-0002-0691-794X
Niko A Busch http://orcid.org/0000-0003-4837-0345

### Ethics
Human subjects: All participants gave written informed consent prior to all experimental procedures. The study was approved by the local ethics committee of the University of Muenster (approval ID 2018-068-f-S) and complied with the Declaration of Helsinki.

### Decision letter and Author response
Decision letter https://doi.org/10.7554/eLife.70907.sa1
Author response https://doi.org/10.7554/eLife.70907.sa2

## Additional files

### Supplementary files
• Transparent reporting form

### Data availability
The anonymised data supporting the findings of this study are openly available from on the Open Science Framework (https://osf.io/ajuzh/).

The following dataset was generated:

| Author(s) | Year | Dataset title | Dataset URL | Database and Identifier |
|---|---|---|---|---|
| Kluger D | 2021 | Respiration aligns perception with neural excitability | https://osf.io/ajuzh/ | Open Science Framework, ajuzh |

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
