## [Editor Report]

Kluger and colleagues investigated the influence of respiration on visual sensory perception in a near-threshold task and argue that the detected correlation between respiration phase and detection precision is liked to α power, which in turn is modulated by the phase of respiration. The main finding is that the moment-to-moment relationship between excitability and perception in turn is coupled to the body's slower respiratory oscillation. This advances our understanding of how the brain-body system works as a whole.

---

## [Decision Letter]

**Decision letter after peer review:**

Thank you for submitting your article "Respiration aligns perception with neural excitability" for consideration by *eLife*. Your article has been reviewed by 3 peer reviewers, and the evaluation has been overseen by a Reviewing Editor and Floris de Lange as the Senior Editor. The following individuals involved in review of your submission have agreed to reveal their identity: Andreas Draguhn (Reviewer #3).

Essential revisions:

Above and beyond the major points of revision raised in the individual reviews, we have agreed on the following queries to be addressed by new analyses (and/or new data, although we are not asking for new data as a hard criterion, obviously, given the current situation):

i) the paper needs more evidence for causality, rather than correlation, including data supporting the specificity of effects for α (compared to other frequencies).

See detailed comments by the Reviewers regarding these issue below.

As an editorial hint, here the authors might want to look into state-of-the-art statistical approaches to inferring causality from observational/correlational data (for review see e.g. https://doi.org/10.1038/s41562-018-0466-5).

At the very least, the question of causality should at least be discussed more explicitly and all causal or causality-insinuating language should be avoided where not warranted.

ii) While we will not ask for new data as a hard requirement in the face of the curren pandemic, the authors should take note of our general concern about the lack of insights in underlying mechanisms; the reviewers feel the authors should at least consider interventions like mouth versus nasal breathing. This would also contribute substantially to the causality-related questions.

iii) better acknowledgement of earlier efforts and findings re. α and breathing and breathing and visual perception is in order. See comments esp. by Reviewer 2 below.

*Reviewer #1 (Recommendations for the authors):*

1. As described above in the public review, it seemed to me to be assumed that respiration causes fluctuations in α power, and fluctuations in α power in turn cause fluctuations in behavior (or we can speak of excitability, and maybe α power is a reflection of that, I'm not hung up on this). Could it be that respiration and α are both fluctuating on their own, and both influence behavior, but the two signals are not connected? I realize that the phase-amplitude coupling analysis is supposed to answer this question. However, this analysis seems problematic in that activity in frequency bands spanning 5-40 Hz (theta to low γ) is coupled to respiration. So there's nothing specific about α/excitability that is related to respiration. Is it possibly the case that "measurability" (signal-to-noise ratio) is rather respiration coupled, as opposed to neural excitability? So, brain activity for whatever reason is easier to measure at certain phases of the respiration cycle? And therefore the brain-behavior relationship is clearer or stronger? That would be boring, but I think important to discuss / rule out. One possibility for the α-causality issue would be some kind of causal or moderator analysis. I'm not an expert, but I think there are some exciting new statistical techniques that could be useful here. I'm less sure about how to better assess the α-specificity of the effect, or to conceptually deal with an unspecific effect.

2. I was not able to understand the analysis that involved shifting the α time course. Unfortunately, I'm not sure what to recommend here, because I really didn't understand well enough to advise on how to make it clearer. I wonder whether a didactic analysis figure panel as part of Figure 4 might help?

3. I think there are a few ways to update the analyses of the psychometric thresholds to make the results more interpretable and also to be more parallel across sections.

– First, the authors use the sine and cosine of a circular predictor in their regression. They then interpret the β values and statistics separately for the two predictors. I think the more common and meaningful approach is to recombine the betas into a single value that reflects the predictive power of the phase predictor, and then calculate corresponding statistics for the combined value. To do so, take the root-mean-square of the b1 and b2 values, sqrt(b1^2^ + b2^2^). Since this value will always be positive, it is often tested for significance using a permutation-based approach, which the authors are already comfortable with. I recently learned that this approach is called "harmonic regression" (for googleableness).

– The approach I described above to recombine betas for the phase predictor will not work in the first analysis the authors present based on mutual information. Actually, I was wondering why the authors chose this approach, when a logistic regression would have worked equally well, would have been more cohesive with the other analyses in the paper, and would afford the same model-comparison approach as they used for the other analysis, instead of the Wilcoxon test on MI values. I would suggest that the results would be more streamlined with a regression approach here that algins with the other sections.

– It might also be nice to substitute the hits > misses contrast on α power with a regression approach examining how α power changes psychometric thresholds so that the analyses start to align better across sections.

4. I think I didn't fully follow the logic in the Intro/Discussion regarding the one of the paths by which respiration might affect excitability. In particular, for the route via the olfactory bulb, it wasn't clear to me how phase-amplitude coupling between slow olfactory, respiration-coupled phase and amplitude of "faster oscillations" would structurally or functionally lead to fluctuations in amplitude of parieto-occipital α oscillations. Is this known, or is this rather for future work?

5. The conversion of the normalized respiration time course to phase involved using Matlab's findpeaks function, assigning peaks and troughs values of 0 and {plus minus} π, respectively, and then linearly interpolating the phase values between them. This seems odd, especially as the later analysis (p. 25) accomplished the same thing via the Hilbert transform (more what I was expecting). I also would ask how, given that values between peak and trough were interpolated, it would be possible to see phase slips and nonlinear phase transitions, as in Figure 1d.

*Reviewer #2 (Recommendations for the authors):*

The study should include a comparison between nasal and oral breathing with reliable monitoring. Video alone might not suffice.

The analysis of the MEG signal should be extended to the entire frequency spectrum captured by the method.

If blood CO2 is the suspected mediator between breathing cycle and excitability the authors should consider manipulations that alter this parameter, using for example, controlled shallow vs. deep breathing.

*Reviewer #3 (Recommendations for the authors):*

Specific recommendations (not sorted by priority):

i) Introduction, lines 75-77: The sentence is a bit puzzling. It starts with "The bidirectionality…", but this bidirectionality had not been introduced previously to the reader. Rather, the connections mentioned in the second part of the sentence may lead to / explain such bidirectional relations.

ii) The raincloud plot in Figure 1c is not easily understandable to unfamiliar readers. Please explain in some detail and, if possible, label x-dimension.

iii) Figure 2c: Can you please provide an additional panel with extended contrast resolution (x-axis) to illustrate the dispersion of the curves more clearly? At present, density of curves and lacking contrast make it almost impossible to see the details.

iv) Figure 3b shows an apparent strong performance-dependent difference in α power AFTER stimulus presentation. This should be made explicit and discussed. Could it be that the difference in this phase of the experiment does also affect the behavioral decision which, obviously, comes after presentation?

v) Figure 4c does also need a more thorough explanation to be understandable.

---

## [Author Response]

Reviewer #1 (Recommendations for the authors):1. As described above in the public review, it seemed to me to be assumed that respiration causes fluctuations in α power, and fluctuations in α power in turn cause fluctuations in behavior (or we can speak of excitability, and maybe α power is a reflection of that, I'm not hung up on this). Could it be that respiration and α are both fluctuating on their own, and both influence behavior, but the two signals are not connected? I realize that the phase-amplitude coupling analysis is supposed to answer this question. However, this analysis seems problematic in that activity in frequency bands spanning 5-40 Hz (theta to low γ) is coupled to respiration. So there's nothing specific about α/excitability that is related to respiration. Is it possibly the case that "measurability" (signal-to-noise ratio) is rather respiration coupled, as opposed to neural excitability? So, brain activity for whatever reason is easier to measure at certain phases of the respiration cycle? And therefore the brain-behavior relationship is clearer or stronger? That would be boring, but I think important to discuss / rule out. One possibility for the α-causality issue would be some kind of causal or moderator analysis. I'm not an expert, but I think there are some exciting new statistical techniques that could be useful here. I'm less sure about how to better assess the α-specificity of the effect, or to conceptually deal with an unspecific effect.

The reviewer touches on several important topics here, namely a) potential confounds or nonspecific (‘boring’) alternative explanations for the reported effects, b) the question of causality in body-brain interactions, and c) relatedly, the specificity/exclusivity of α modulations.

To start with the latter, the reviewer rightfully states that the respiration-related modulation effects we report are not restricted to the α band. In fact, we have shown in a previous study (Kluger and Gross, PLoS Biol 2021) that the pattern of respiration-modulated brain oscillations (RMBOs) is spatially and spectrally organised throughout the entire brain. Therefore, we do not want to claim at all that respiration exclusively modulates α oscillations. As oscillations across the frequency band have been implicated in a wide variety of perceptual and cognitive tasks, we a priori selected α power as an excitability marker based on a rich background of previous studies in visual perception research (kindly see the manuscript for references).

The reviewer then asks whether respiration and α independently influence behaviour *without being connected*, which we can confidently negate. Both present and previous findings from our group and others clearly show that α power cyclically changes over the respiration cycle, which is the precise purpose of phase-amplitude coupling analyses. A far less trivial (but equally justified) question is whether the pattern of results we observe here is sufficient to assume a causal link, which we should have discussed more cautiously. Unfortunately, ‘causal’ analysis methods like directed connectivity measures (e.g., Granger causality, transfer entropy, …) or DCM are not an option for the present study, as these analyses quantify the (directed) interactions between two continuous time series, whereas the behavioural criterion we investigate consists of discrete events. That being said, we acknowledge that there are conceivable mechanisms we cannot rule out (such as an unknown variable driving both the respiration-α and respiration-behaviour effects), and we have substantially revised our manuscript in this regard (see below).

Thus, although we do not know the precise causal mechanism underlying our findings, it is important to consider the evidence *against* several artificial or nonspecific effects. The reviewer suggests that fluctuations in ‘measurability’ (i.e., SNR changes over the respiration cycle) could contribute to our results. It is certainly true that respiration induces time-locked head movements (albeit very small ones), which translate to slight changes of head position relative to the MEG sensors. In previous work (Kluger and Gross, NeuroImage 2020; Kluger and Gross, PLoS Biol 2021), we have quantified these effects and shown that the comprehensive removal of head movement artefacts (by means of a 36-factor regression model and built-in motion tracking that we also employed in the current study, see Methods) effectively controls for respiration-related SNR changes (for the original description of the method, kindly see Stolk et al., NeuroImage 2013). Furthermore, as the reviewer points out, any SNR-based explanation would imply that high SNR leads to a stronger/clearer body-brain interaction. In fact, we describe the opposite effect, namely a link between *α suppression* and behaviour.

These considerations notwithstanding, we have revised our manuscript in response to the reviewer’s remarks. Specifically, we have phrased the interpretation of our results more cautiously and now discuss the question of causality explicitly:

“The bootstrapping procedure yielded a confidence interval of [-33.17 -29.25] degrees for the peak effect of α power. While these results strongly suggest that respiration-α coupling temporally precedes behavioural consequences, they do not provide sufficient evidence for a strict causal interpretation (see Discussion).” (p. 11)

“Rigorous future work is needed to investigate potentially causal effects of respiration-brain coupling on behaviour, e.g. by means of directed connectivity within task-related networks. A second promising line of research considers top-down respiratory modulation as a function of stimulus characteristics (such as predictability). This would grant fundamental insights into whether respiration is actively adapted to optimise sensory sampling in different contexts, as suggested by the animal literature.” (p. 15)

2. I was not able to understand the analysis that involved shifting the α time course. Unfortunately, I'm not sure what to recommend here, because I really didn't understand well enough to advise on how to make it clearer. I wonder whether a didactic analysis figure panel as part of Figure 4 might help?

We are grateful for the reviewer’s comment and their suggestion to add a schematic, didactic panel to illustrate the analysis shown in Figure 4. We gladly adopted their suggestion and have restructured Figure 4 to now include such a panel. In addition, we have revised the Methods and Results section and the figure legend in order to make the underlying analysis more accessible. We appreciate any further suggestions or remarks the reviewer might have.

3. I think there are a few ways to update the analyses of the psychometric thresholds to make the results more interpretable and also to be more parallel across sections.– First, the authors use the sine and cosine of a circular predictor in their regression. They then interpret the β values and statistics separately for the two predictors. I think the more common and meaningful approach is to recombine the betas into a single value that reflects the predictive power of the phase predictor, and then calculate corresponding statistics for the combined value. To do so, take the root-mean-square of the b1 and b2 values, sqrt(b1^2^ + b2^2^). Since this value will always be positive, it is often tested for significance using a permutation-based approach, which the authors are already comfortable with. I recently learned that this approach is called "harmonic regression" (for googleableness).

We thank the reviewer for their suggestion regarding the joint interpretation of respiratory sine and cosine effects. In response to their remarks, we have adapted our LMEM computations accordingly. Thus, we now combine the LMEM weights for sine and cosine of respiration into one ‘phase vector norm’ and test the significance of these combined weights by means of a permutation approach. We have rewritten both the Methods and Results sections and adapted Figures 2 and 4 to include a display of the corresponding results.

“We set up a linear mixed effect model (LMEM) expressing the re-fitted PsychF threshold values as a linear combination of sine and cosine of the respiration signal. The resulting regression weights for sine and cosine were combined in a phase vector norm (akin to a harmonic regression) and tested for significance using 5000 random permutations of subject-level PsychF threshold courses (see Methods for details). The empirical regression weight was greater than any one from the randomisation distribution (see Figure 2d), corroborating the overall influence of respiration on perceptual performance.”(p. 6)

“To assess the significance of α contributions to the interplay of respiration and performance, we set up a second linear mixed effect model (LMEM) expressing re-fitted PsychF threshold values as a linear combination of sine and cosine of the respiration signal as well as parieto-occipital α power (averaged from 8 – 13 Hz). […] The resulting β weights for sine and cosine were combined as described above and saved as ‘null vector norms’. Significance of the empirical vector norm was determined by computing its percentile rank relative to the density function of the null distribution.” (p. 26)

“In keeping with our previous LMEM analyses, we also generated a null distribution for the empirical phase vector norm at each lag by randomly shuffling each participant’s PsychF threshold vector n = 1000 times. These null distributions are shown as violin plots in Figure 4e.” (p. 27)

– The approach I described above to recombine betas for the phase predictor will not work in the first analysis the authors present based on mutual information. Actually, I was wondering why the authors chose this approach, when a logistic regression would have worked equally well, would have been more cohesive with the other analyses in the paper, and would afford the same model-comparison approach as they used for the other analysis, instead of the Wilcoxon test on MI values. I would suggest that the results would be more streamlined with a regression approach here that algins with the other sections.

We appreciate the reviewer’s suggestion of a logistic regression analysis. We opted for the mutual information approach due to its sensitivity for non-linear effects (which the logistic regression would not have picked up), since this analysis was to provide a very first look into the link between respiration and behaviour. Based on these initial results, subsequent LMEMs were used to characterise the involvement of α power (see Figure 2) and its modulation over different respiratory phase lags (see Figure 4).

– It might also be nice to substitute the hits > misses contrast on α power with a regression approach examining how α power changes psychometric thresholds so that the analyses start to align better across sections.

At this point, we felt it was critical to investigate all ‘pairwise’ relationships (i.e., including α power <> behaviour) as a precondition for the triadic interplay of respiration, oscillations, and behaviour. Therefore, it is important to note that the contrast of hits vs misses was not intended to characterise α and dynamic threshold changes in the first place – this was the next step, for which we employed LMEM (shown in Figure 4). Before that, as we state in the manuscript (kindly see p. 9), we aimed for a more fundamental replication of well-described α suppression effects (see Figure 3). Since these ‘classical’ findings almost exclusively stem from contrasts of hits vs misses, we decided to follow the same approach for the sake of comparability.

4. I think I didn't fully follow the logic in the Intro/Discussion regarding the one of the paths by which respiration might affect excitability. In particular, for the route via the olfactory bulb, it wasn't clear to me how phase-amplitude coupling between slow olfactory, respiration-coupled phase and amplitude of "faster oscillations" would structurally or functionally lead to fluctuations in amplitude of parieto-occipital α oscillations. Is this known, or is this rather for future work?

The reviewer is completely correct in suspecting that the precise mechanism underlying respiration-excitability coupling is not yet fully understood. In fact, while recent work in human perception and cognition has accumulated a sizable amount of evidence for a link between respiration and behaviour, the present study is the first to suggest excitability as a driver of these known effects. That being said, our proposal of excitability involvement brings together two well-established lines of research, namely i) the tight link between α suppression, neural excitability, and (visual) perception in both animal and human studies, and ii) the fundamental mechanism of respiration-induced rhythms in the olfactory tract (i.e., epithelium and olfactory bulb) being translated to neural oscillations of higher frequencies. The latter greatly relies on animal work, as non-invasive measurements of human bulbar activity are just being developed (see Iravani et al., Nat Comms 2020). These animal studies have clearly demonstrated the first step of the coupling cascade, namely how, during nasal inspiration, air enters the respiratory tract and triggers mechanoreceptors connected to the olfactory bulb. This initiates infraslow neural oscillations closely following the respiratory rhythm (*phase-phase coupling*). The phase of these slow neural oscillations then drives the amplitude of faster oscillations (*phase-amplitude coupling*) and propagates to upstream areas both within and beyond the olfactory system. We have shown in previous resting-state work that respiration-related modulations of oscillatory power i) are widespread across cortical and subcortical brain areas, and ii) differentially affect frequencies across the entire spectrum (see Kluger and Gross, PLoS Biol 2021). From this perspective, the present findings of parieto-occipital changes in α power are not surprising and fit very well with the functional significance ascribed to α oscillations in visual perception (as cited in the main text). Furthermore, even though the mechanisms behind bulbo-cortical propagation in humans are difficult to assess directly, there is convincing evidence from multiple studies (e.g., Zelano et al., JNeurosci 2016; Perl et al., Nat Hum Behav 2019) that both respiration-related power changes and behavioural effects dissipate when participants breathe through the mouth instead of the nose. Zelano and colleagues (2016) were able to show that oral breathing not only circumvents the (apparently necessary) mechanosensory stimulation within the olfactory tract, but also leads to disorganisation of slow-oscillation phase synchrony within the limbic system. As we state in the Discussion, these slow neural oscillations – paced by rhythmic nasal respiration – are part of a nested organisation of neural oscillations (Lakatos, 2005) and serve as a carrier rhythm for higher-frequency oscillations throughout the brain.

In order to clearly communicate the status quo of understanding these mechanisms in humans, we have amended the Discussion section as follows:

“The authors later proposed two main sources of respiration-locked neural signalling, namely the olfactory bulb (OB) and extrabulbar sources within the brainstem (Heck et al., 2019). As outlined in the introduction, there is broad consensus that cross-frequency coupling (Canolty and Knight, 2010; Jensen and Colgin, 2007) plays a central role in translating respiratory to neural rhythms: Respiration entrains neural activity within the olfactory tract via mechanoreceptors, after which the phase of this infraslow rhythm is coupled to the amplitude of faster oscillations (see Fontanini and Bower, 2006; Ito et al., 2014). While this mechanism is difficult to investigate directly in humans, converging evidence for the importance of bulbar rhythms comes from animal bulbectomy studies (Ito et al., 2014) and the fact that respiration-related changes in both oscillatory power and behaviour dissipate during oral breathing (Zelano et al., 2016; Perl et al., 2019). Thus, rhythmic nasal respiration conceivably aligns rhythmic brain activity across the brain, which in turn influences behaviour. In our present paradigm, transient phases of heightened excitability would then be explained by decreased inhibitory influence on neural signalling within the visual cortex, leading to increased postsynaptic gain and higher detection rates. Given that the breathing act is under voluntary control, the question then becomes to what extent respiration may be actively used to synchronise information sampling with phasic states of heightened excitability. “(p. 13 ff)

5. The conversion of the normalized respiration time course to phase involved using Matlab's findpeaks function, assigning peaks and troughs values of 0 and {plus minus} π, respectively, and then linearly interpolating the phase values between them. This seems odd, especially as the later analysis (p. 25) accomplished the same thing via the Hilbert transform (more what I was expecting). I also would ask how, given that values between peak and trough were interpolated, it would be possible to see phase slips and nonlinear phase transitions, as in Figure 1d.

Thank you for giving us the opportunity to clarify our methodological approach with regard to extraction of the respiratory phase. While the Hilbert transform is indeed a common approach for phase extraction, its applicability for respiratory data is limited. The reason is that i) the shape of the respiratory signal itself is not symmetric (i.e. there are differences in the ‘raw’ duration of inhalation and exhalation) and ii) this asymmetry varies both within and (to a greater extent) between participants. The Hilbert transform does not consider these deviations from a pure sinusoid shape, which is why – over the course of multiple studies – we found the two-point interpolation to be a more accurate method for phase extraction. This interpolation is well-grounded in dynamical systems theory (see e.g. ‘Synchronization’ by Pikovsky, Rosenblum and Kurths, Cambridge U Press, 2010) and allows us to employ between-subject comparisons based on a more realistic shape of the respiratory signal.

We would like to note that we are currently preparing a tutorial-style manuscript where we outline methodological considerations (such as phase extraction) and compare parameter settings for handling respiratory data. As for the present manuscript, we certainly agree with the reviewer that a consistent use of one phase extraction method would be more intuitive, which is why we re-conducted the analyses of modulation index (MI) and phase-triggered average (PTA) shown in Figure 4. While the absolute MI values slightly decreased, the overall pattern of results remained identical: As in the previous analysis, all frequencies between 2 – 40Hz were significantly modulated by respiration.

Reviewer #2 (Recommendations for the authors):The study should include a comparison between nasal and oral breathing with reliable monitoring. Video alone might not suffice.

This closely relates to the reviewer’s previous comment regarding well-established differences in respiration-brain coupling during nasal vs oral breathing. The reviewer rightfully noted that there is now considerable evidence showing that, by simply circumventing mechanosensory stimulation within the olfactory epithelium and olfactory bulb, oral breathing does not induce the kind of effects we report here (kindly see our response above for more details). Therefore, the consistency of our results pattern with previous work lends further credibility to our continuous monitoring of participants’ breathing, as we would not have observed the present effects had participants indeed breathed orally.

The analysis of the MEG signal should be extended to the entire frequency spectrum captured by the method.

In response to the reviewer’s detailed remarks above, we have amended the manuscript to now contain control analyses for the entire frequency spectrum (kindly see above).

If blood CO2 is the suspected mediator between breathing cycle and excitability the authors should consider manipulations that alter this parameter, using for example, controlled shallow vs. deep breathing.

We thank the reviewer for their insightful comment. As outlined above, we have removed the Discussion paragraph on CO2 blood levels, as we no longer believe this to be a core mechanism of the observed effects. As a side note, for a comparison of automatic vs deep breathing, we would like to refer the reviewer to previous work from our lab (Kluger and Gross, NeuroImage 2020).

Reviewer #3 (Recommendations for the authors):Specific recommendations (not sorted by priority):i) Introduction, lines 75-77: The sentence is a bit puzzling. It starts with "The bidirectionality…", but this bidirectionality had not been introduced previously to the reader. Rather, the connections mentioned in the second part of the sentence may lead to / explain such bidirectional relations.

Thank you for spotting the poor phrasing in the Introduction section. The respective paragraph has been rephrased and now reads as follows:

“Despite being largely automatic, breathing can also be adapted top-down when required, for example during speech or laughter (McKay et al., 2003). To this end, intricate connections from key structures like the preBötzinger complex (through the central medial thalamus) and the olfactory bulb to both the limbic system (Carmichael et al., 1994) and the neocortex (Yang and Feldman, 2018) serve the bidirectional interplay between respiratory control and higher cognitive functions.”(p. 3)

ii) The raincloud plot in Figure 1c is not easily understandable to unfamiliar readers. Please explain in some detail and, if possible, label x-dimension.

We thank the reviewer for requesting further clarification of Figure 1. Following their suggestions, we have amended the figure legend to provide further details. As the ‘cloud’ section of the illustration merely shows a probability density function, it is conventional not to label this axis in group comparisons (kindly see e.g. the original method paper by Allen et al., PeerJ 2018). In order to avoid any misunderstandings, we have included this information in the figure legend as well:

“Figure 1. Task and behavioural results. a, In the experimental task, participants kept their gaze on a central fixation cross while a brief, near-threshold Gabor patch (magnified for illustrative purposes) was randomly presented either on the left or the right side of the screen. During catch trials, no stimulus was presented. After a brief delay, participants were prompted to indicate whether they had seen a stimulus on the left (index finger) or on the right side (middle finger), or no stimulus at all (thumb). b, Exemplary segment of respiration recordings (normalised, top) plotted against its phase angle (bottom). Over time, targets were randomly presented over the respiration cycle and could be either detected (*hits*, green) or undetected (*miss*, yellow). c, Raincloud plots show mutual information between task performance and stimulus contrast (left) and stimulus contrast plus respiration phase (right), respectively. Each dot for *contrast* or *contrast + phase* represents one participant’s mutual information value computed across all trials for the respective condition. Links between dots identify mutual information values from the same participant and illustrate the increase in mutual information when respiration phase is included in its computation: Mutual information was significantly enhanced by adding respiration phase (Wilcoxon signed rank test: *z* = 4.78, *p* < .001). Filled curves represent the respective probability density functions of mutual information values across participants. Raincloud plots generated with the *RainCloudPlots* toolbox for Matlab (Allen et al., 2019b).”

iii) Figure 2c: Can you please provide an additional panel with extended contrast resolution (x-axis) to illustrate the dispersion of the curves more clearly? At present, density of curves and lacking contrast make it almost impossible to see the details.

We appreciate the reviewer’s suggestion to show the dispersion of PsychF thresholds more clearly and have added a new inset visualising threshold at greater detail. In revising the figure, we noticed that it was probably not immediately obvious that panel c (just like panels a-b) illustrated our approach by means of data from a single exemplary participant. We apologise for any confusion this may have caused and have revised the figure legend accordingly.

For a more transparent illustration of PsychF threshold variance across participants, we have added a new Figure 2 —figure supplement 2. Here, we show bin-wise PsychF thresholds (grey) and the overall fit (black) for each participant (as shown in Figure 2c).

iv) Figure 3b shows an apparent strong performance-dependent difference in α power AFTER stimulus presentation. This should be made explicit and discussed. Could it be that the difference in this phase of the experiment does also affect the behavioral decision which, obviously, comes after presentation?

Thank you for this observation, as it allows us to build on a crucial point we did not discuss explicitly enough in our previous manuscript.

Post-stimulus differences in α power have often been observed in M/EEG studies involving “seen/unseen” responses in near-threshold detection tasks (Wyart and Tallon-Baudry, 2009; Bareither et al., 2014; Iemi et al., 2017), and our results are well in line with these findings. One reason for this strong stimulus-locked effect is that perceived stimuli are associated with a stronger evoked response, compared to the one elicited by undetected stimuli (Busch and VanRullen, 2010; Bareither et al., 2014). This evoked response we observe as high power at low frequencies in Figure 3b, is associated with an event-related desynchronization of α, leading to increase in α power for hits vs misses.

The question raised here, namely whether power effects on performance are dependent on the prestimulus or on the post-stimulus time window, is of fundamental importance, and it was already targeted by Wyart and Tallon-Baudry (2009). In this seminal work, the authors observed that both prestimulus and post-stimulus variations in α band power were associated with seen/unseen decisions in a detection task analogous to the one used in our present study. In a subsequent modelling analysis, they showed that spontaneous fluctuations in α band, in their case associated to the locus of spatial attention, influenced indirectly the subsequent model-based decision via its correlation with the post-stimulus α-band activity.

Given the striking similarities between the pattern of results presented here and the ones reported by these previous works, we suggest that the same mechanism is in action here: Spontaneous variations in α-band power preceding the stimulus onset influence the further α decrease associated with a “seen” report (for hits), or an increase leading to an “unseen” report (for misses). The additional information we provide with the current evidence is that these states of prestimulus α power are coupled to specific phases in the respiration cycle. Thus, we consider our findings as evidence for a new factor modulating perceptual decision making processes.

In response to the reviewer’s comment, we have amended the Results section for an explicit discussion of post-stimulus α effects:

“In line with previous work (Wyart and Tallon-Baudry, 2009; Iemi et al., 2017), we further observed a pronounced α desynchronisation after stimulus onset for detected (vs undetected) targets. Such differences have been shown to be driven by stronger evoked responses for these stimuli in near-threshold perception tasks (Busch and VanRullen, 2010; Bareither et al., 2014) and to influence detection performance together with prestimulus α fluctuations (Wyart and Tallon-Baudry, 2009).”(p. 8)

v) Figure 4c does also need a more thorough explanation to be understandable.

Thank you for giving us the opportunity to provide further explanation regarding Figure 4. In response to suggestions from Reviewer 1 (kindly see above), this figure now includes a new didactic panel to clarify both, our methodological approach and its results.